# Acute Effects of the Interval and Duration of Intermittent Exercise on Arterial Stiffness in Young Men

**DOI:** 10.3390/ijerph192416847

**Published:** 2022-12-15

**Authors:** Jianchang Ren, Haili Xiao, Ping Wang

**Affiliations:** College of Physical Education and Sports Science, Lingnan Normal University, Zhanjiang 524048, China

**Keywords:** intermittent exercise, continuous exercise, arterial stiffness, interval

## Abstract

We proved the hypothesis that intermittent exercise would have a better effect on arterial stiffness by shortening the duration of intervals and increasing the number of bouts. Twenty healthy male college students (20.4 ± 0.4 years) were randomly assigned to a quiet control (CON), 30 min continuous exercise (CE), long-interval intermittent exercise with long intervals (IELL), long-interval intermittent exercise with short intervals (IELS), and short-interval intermittent exercise with short intervals (IESS). The intensity was set to 45% of the heart rate reserve. The brachial-ankle pulse wave (baPWV) was measured at baseline (BL), 0 min post-exercise, 20 min post-exercise, 40 min post-exercise, and 60 min post-exercise. BaPWV changes (⊿baPWV) from the BL in the same tests were used for the analysis. ⊿baPWV did not change significantly in the CON. ⊿baPWV decreased significantly at 0, 20, and 40 min in all exercise tests. ⊿baPWV decreased significantly at 60 min in IELS and IESS. At 60 min, the ⊿baPWV of IELS and IESS was still significantly lower than that of CON and CE, and the ⊿baPWV of IESS was still significantly lower than that of IELS. Hence, shortening the intervals of intermittent exercise and increasing the number of repetitions may enhance the effect of improving arterial stiffness.

## 1. Introduction

Cardio-cerebrovascular disease (CVD), represented by strokes and heart attacks, is increasingly occurring in young people [1]. As a non-communicable chronic disease, the early control of CVD risk factors can effectively prevent its occurrence. Arterial stiffness is valuable in predicting cardiovascular disease risk. Increased arterial stiffness is an independent cardiovascular disease risk factor, and physical activity may have a preventive effect on cardiovascular disease by improving arterial stiffness [2]. Several parameters can be used to evaluate the hardness of human arteries—for example, pulse wave velocity (PWV); an increase in PWV represents a decrease in vessel wall elasticity, distensibility, and compliance and an increase in stiffness [3]. Cohort studies have demonstrated that aortic PWV predicts cardiovascular all-cause mortality [4]. Brachial-ankle pulse wave (baPWV) conduction velocity is a systemic arterial stiffness measurement method measured by analyzing brachial and tibial arterial waves [5]. BaPWV is considered the “gold standard” for evaluating arterial stiffness [6]. According to Mitchell et al. [7], higher arterial stiffness is associated with an increased risk of first cardiovascular events. Over 10–12 years, a 1 m/s increase in baPWV was associated with a 0.9% absolute risk of stroke, a 2.2% risk of death, and a 1.4% risk of coronary heart disease. According to research, increased arterial stiffness in the elderly can harm brain health [8]. Arterial stiffness is caused by aging, hypertension, and inflammation [9].

Regular physical activity can improve health and prevent cardiovascular disease [10]. Many studies have shown that exercise reduces cardiovascular risk, morbidity, and mortality [11,12]. Aerobic exercise can help individuals improve their cardiorespiratory fitness and their vascular health. Increased aerobic capacity, for example, lowers blood pressure and arterial stiffness. Regular exercise improves arterial stiffness, long-distance running helps reduce arterial stiffness [13], low-intensity resistance exercise helps reduce arterial stiffness [14], and high-intensity intermittent exercise helps reduce arterial stiffness in obese women [15]. People who exercise prefer moderate-intensity interval exercise to high-intensity continuous exercise and interval exercise because it is more acceptable and easier to stick to. The American College of Sports Medicine guidelines recommend continuous and intermittent exercise to stay healthy [16]. Regular exercise can improve arterial stiffness. For instance, Hu et al. [11] reported that, after four weeks, the arterial stiffness of female university students decreased compared with that before moderate-intensity aerobic exercise training. Previous studies have shown that intermittent and continuous exercise can improve arterial stiffness. Zheng et al. [17] showed that acute intermittent exercise is more beneficial than continuous exercise in reducing arterial stiffness for the same amount of exercise. However, the effect of intermittent exercise disappears when the intervals of intermittent exercise are too long [17], while the effect of intermittent exercise can be maintained by increasing the number of bouts of intermittent exercise [18]. These findings suggest that the duration and number of sessions affect the acute effects of intermittent exercise on arterial stiffness. Intermittent exercise is more acceptable to most people than continuous exercise. However, few studies have found the optimal interval duration and number of bouts.

A portion of young people lack the habit of exercise due to high work demands and limited time, but the atherosclerotic process starts during the stage of healthy adolescence [19]. Therefore, designing accessible and feasible exercise prescriptions to reduce arterial stiffness for an audience of healthy young adults is important to improving their vascular health and preventing CVD at an early stage. Hence, to investigate the superiority of intermittent exercise on the improvement of arterial stiffness, we hypothesized that intermittent exercise would have a better effect on arterial stiffness by shortening the duration of intervals and increasing the number of bouts.

## 2. Materials and Methods

### 2.1. Subject

Twenty healthy male college students aged 20.4 ± 0.4 years were enrolled in the experiment (Table 1). Before participating in the experiment, participants filled out an informed consent form. All subjects were in good health, were non-smokers, and were free of any disease affecting the cardiovascular system. A medical health examination was completed by all subjects. This study was conducted in a manner consistent with the Declaration of Helsinki and was authorized by the Ethics Committee of Lingnan normal university (LNU20210501). All participants were free of strenuous exercise and alcohol or coffee intake for three days before the trial. The subjects arrived at the laboratory in the morning. Fasting was maintained, and quiet rest was taken for at least 30 min prior to baseline measurements. All experiences were conducted in the exercise and cardiovascular health laboratory with a room temperature of 23–26 degrees.

### 2.2. Study Design

The balanced cross-self-controlled design was adopted, and all participants enrolled in five tests in randomized order, with each test separated by seven days. The five tests included the control (CON), a 30 min continuous exercise trial (CE), long-interval intermittent exercise with long intervals (IELL, two repeats of 10 min cycling sessions with 10 min pauses between sessions), long-interval intermittent exercise with short intervals (IELS, two repeats of 10 min cycling sessions with a 5 min pause between sessions), and short-interval intermittent exercise with shorts intervals (IESS, four repeats of 5 min cycling sessions with a 5 min pause between sessions), as seen in Figure 1. All experiments were conducted from 7:00 am to 11:30 am. At baseline (BL) and at 0, 20, 40, and 60 min after exercise, respectively, baPWV was evaluated. In the exercise test, subjects performed moderate-intensity exercise (pedal frequency of 60–70 revolutions per minute) on a power bike.

To observe changes in baPWV after each interval in the IESS experimental group, three additional measurements of baPWV were performed before the second (B1), third (B2), and fourth (B3) 5 min exercise periods, as shown in Figure 1 for the IESS test.

### 2.3. Exercise Protocol

The exercise tests were performed on an electromagnetic brake bicycle dynamometer (Ergoselect 100P, Ergoline GmbH, Bitz, Germany). The intensity was set to 45% of the heart rate reserve using the following formula: target heart rate = [220 − age − resting heart rate] × 45% (45% of exercise intensity) + resting heart rate (using the Kavonen formula). During exercise, the exerciser’s heart rate was monitored by a heart rate band sensor paired with the power bike. The loading load is manually adjusted to maintain the exerciser’s target heart rate.

### 2.4. Arterial Stiffness Measurement

The baPWV measure of arterial stiffness and blood pressure was collected bilaterally through the brachial and posterior tibial (ankle) arteries using the BP-203RPE II (Omron Colin, Tokyo, Japan) after 10 min of quiet rest. The instrument operator was specially trained, the subject lay flat on the test bed while sterile alcohol was wiped on the left upper arm, right upper arm, left ankle, and right ankle, and the appropriate wrist strap was fitted. ECG clips were placed on both the left and right wrists. During the test, a microphone was used to monitor the heart sounds (cardiogram). After the electrocardiogram was stable, heart sounds were detected, and “PCG OK” was displayed; press the “Start” button, and the system will automatically obtain the left and right baPWV values without manual intervention. The average baPWV was calculated for the left and right sides, and their change from the baseline in the same experiment (⊿baPWV) was calculated and used for later analysis. Blood pressure and heart rate were also measured with the BP-203RPE II.

### 2.5. Statistical Analysis

Data are expressed as the means ± SE. Changes in ⊿baPWV across exercise protocols were analyzed by four independent two-factor repeated-measures ANOVAs in a three (group) × four (time) format. Significant variations across trials were identified using the Bonferroni post hoc test. The assumption of sphericity was tested using the Mauchly sphericity test to determine whether it was true. A one-way ANOVA with repeated measurements and Bonferroni post hoc tests were performed to observe the time course of the ⊿baPWV dynamics in the IESS test. The level of statistical significance was set at *p* < 0.05. Origin 2022b was used for the data analysis.

## 3. Results

### 3.1. Subject’s Baseline Characteristics

Table 1 displays the subjects’ initial characteristics, such as age, height, weight, body mass index, and blood pressure. In addition, the subjects had arterial stiffness comparable to those of their actual age according to the BP-203RPE II vascular screening system.

### 3.2. Arterial Stiffness

The mean (±SD) variation of the ⊿baPWV over the five trials is shown in Figure 2 and Figure 3. ⊿baPWV was not changed in the CON trial (0.00 ± 0.00, −3.14 ± 1.52, 3.64 ± 1.34, 10.55 ± 3.21, 6.96 ± 2.17 at BL, 0 min, 20 min, 40 min, and 60 min, respectively). In the CE test, ⊿baPWV varied with time (0.00 ± 0.00, −86.12 ± 10.59, −60.42 ± 15.34, −41.85 ± 10.55, −7.92 ± 11.52 at BL, 0 min, 20 min, 40 min, and 60 min, respectively). In the IELL test, ⊿baPWV varied with time (0.00 ± 0.00, −55.10 ± 12.70, −87.10 ± 11.56, −62.40 ± 9.67, −6.7 ± 2.32 at BL, 0 min, 20 min, 40 min, and 60 min, respectively). In the IELTS test, ⊿baPWV varied with time (0.00 ± 0.00, −103.83 ± 28.93, −55.67 ± 16.60, −49.33 ± 9.08, −28.67 ± 36.51 at BL, 0 min, 20 min, 40 min, and 60 min, respectively). In the IESS test, ⊿baPWV varied with time (0.00 ± 0.00, −62.00 ± 25.50, −108.5 ± 21.50, −103.50 ± 27.00, −83.00 ± 27.50 at BL, 0 min, 20 min, 40 min, and 60 min, respectively).

As shown in Figure 2 and Figure 3, the interaction between the treatment and time reached significance (*p* < 0.01 in Figure 2a, *p* < 0.0001 in Figure 2b; *p* < 0.0001 in Figure 3a,b), and this showed that the time-dependent ⊿baPWV was different in each trial. Additionally, the primary influence of time was significant (*p* < 0.0001 in Figure 2 and Figure 3). It demonstrated that, over time, ⊿baPWV showed significant alterations. There were significant primary consequences of the treatment (*p* < 0.01 in Figure 2a, *p* < 0.0001 in Figure 2b, *p* < 0.001 in Figure 3a, and *p* < 0.0001 in Figure 3b).

The Bonferroni post-test showed that ⊿baPWV was significantly lower in exercise trials than it was in CON at the 0 min time point (* *p* < 0.001, CE, IELL, IELS, and IESS compared with CON; Figure 2 and Figure 3). The ⊿baPWV was significantly lower in the exercise test than it was in the CON at 20 min (^#^
*p* < 0.05, CE, IELL, IELS, and IESS compared with CON; Figure 2 and Figure 3). At 40 min, ⊿baPWV was significantly lower in the exercise test than it was in the CON (@ *p* < 0.05, CE, IELL, IELS, and IESS compared with CON; Figure 2 and Figure 3). At 60 min, the ⊿baPWV of IELS was significantly lower than that of CON (& *p* < 0.05, Figure 2b) and CE (** *p* < 0.05, Figure 2b). At 60 min, the ⊿baPWV of IESS was significantly lower than that of CON (% *p* < 0.05, Figure 3), CE ($$ *p* < 0.05, Figure 3a), and IELS ($ *p* < 0.05, Figure 3b).

Figure 4 shows the change in ⊿baPWV in IESS relative to its baseline. ⊿baPWV in the mean ± SD decreased significantly from 0.00 ± 0.00 at baseline (BL) to −25.55 ± 15.20, −30.23 ± 16.50, −53.00 ± 19.90, −62.00 ± 15.5, −108.5 ± 18.50, −103.50 ± 17.00 and −83.00 ± 17.60 at B1, B2, B3, 0 min, 20 min, 40 min, and 60 min, respectively.

## 4. Discussion

The effect of intermittent and continuous exercise on arterial stiffness has been examined in acute exercise and long-term training [11,12,15]. Our results demonstrated that intermittent exercise with small amounts of exercise improved arterial stiffness at least to the same extent, or even better, compared to continuous exercise with large amounts of exercise. These findings supplement the emerging literature indicating that short-duration rest from intermittent exercise helps reduce the risk of cardiovascular disease.

### 4.1. Effects of Intervals of Intermittent Exercise and Periods of Exercise on Arterial Stiffness

Previous studies have shown that high-intensity intermittent exercise is more beneficial than moderate-intensity continuous exercise for improving arterial stiffness. For instance, Hortman et al. [15] noted that high-intensity interval and moderate-intensity training did not increase cfPWV acutely, and only high-intensity training could reduce the argumentation index (AIx) in obese young women. Likewise, Guimaraes et al. [20] suggested that continuous and internal exercise training benefits blood pressure control, but only interval training reduces arterial stiffness in treated hypertensive subjects. However, given the association between exercise intensity and cardiovascular system response, determining whether the benefits of high-intensity intermittent exercise are due to the high intensity or to the short intervals between exercises can be difficult. Hence, this study considers the amount of continuous and intermittent exercise to be comparable, and the role of intermittent exercise can be isolated.

Preceding studies have shown that two 15 min low-intensity exercises at 20 min apart produce a greater improvement in arterial stiffness than one 30 min bout of continuous exercise of the same intensity [21]. However, most people do not choose this type of exercise because the intervals are too long. The American College of Sports Medicine recommends moderate-intensity exercise for daily physical activity [22]. The present study demonstrated that two 10 min moderate-intensity rides separated by 10 min caused the same degree of improvement in arterial stiffness as one 30 min continuous ride, the latter having more exercise volume. Two 10 min exercises with 10 min intervals in between are comparable to one 30 min exercise (IELL in Figure 1 and Figure 2a). This exercise has the benefit of being more acceptable and has the same effect of improving arterial stiffness.

The advantage of intermittent exercise depends on the previous exercise’s residual effect and disappears when the interval is too long. Zheng et al. [17] showed that two 15 min sessions of moderate-intensity cycling separated by 20 min improved arterial stiffness more than 30 min of continuous exercise of the same intensity. However, the advantage of intermittent exercise over continuous exercise disappears when the interval between exercises is extended to 60 min, which inspired the hypothesis that shortening the interval between exercises may enhance the effect of intermittent exercise. Our findings suggest that two 10 min exercise regimens with a 5 min interval between exercises are superior to one 30 min continuous exercise (IELS in Figure 1 and Figure 2b). Additionally, Ronsen et al. [23] reported that, compared with six hours of rest after the first round of exercise, three hours of rest before the second round will cause more obvious changes in neuroendocrine factors and the white blood cell count. These findings may prove our hypothesis that shortening the interval between exercises is beneficial to enhancing the effects of interval exercise. Moreover, in the IELS trial, the total time for exercise and intervals was reduced to 25 min, which is even shorter than 30 min of continuous exercise, making it the most time-efficient protocol in this study. This exercise protocol is also more accessible.

### 4.2. Effect of the Number of Intervals of Intermittent Exercise on Arterial Stiffness

Although the guidelines recommend 10–15 min of exercise [24], some people cannot sustain continuous exercise for long periods. Therefore, the benefits of exercising for short periods (< or = 5 min) should also be explored. Several studies [25,26] have used 5 min of exercise time in their protocols, demonstrating that the same exercise effect can be produced with very short exercise times.

In this study, we designed a form of intermittent exercise with four 5 min bouts of exercise with an interval of 5 min each time (IESS in Figure 1). Our results suggested that it has a superior effect on arterial stiffness than one 30 min continuous exercise session. However, this is inconsistent with the idea [27] that adults may derive similar health benefits from a single bout of exercise or accumulate activity from shorter bouts of exercise. The advantages can be related to many factors, including the exercise regimen used and the participants involved. Our findings are consistent with a recent study [25] that used a lower exercise intensity. We also compared the improvement in arterial stiffness induced by four 5 min exercises separated by 5 min each and two 10 min exercises separated by 5 min each (IESS and IELS in Figure 1 and Figure 3b).

In contrast, at 60 min, arterial stiffness remained significantly lower with four 5 min intervals of exercise than with IELS, indicating that four 5 min intervals of exercise produced a greater reduction in arterial stiffness than two 10 min intervals of exercise. The alternations in arterial stiffness with exercise (Figure 4) indicated that ⊿baPWV decreased gradually after 5 min of repetitive exercise. This result may be related to the cardiovascular response and hemodynamic changes induced by intermittent exercise [12,28,29,30]. One study [31] showed an increase in the plasma levels of NOx after the second of two consecutive exercise stress tests 24 h apart.

This study has practical implications for multiple physical activities completed within one hour. Reducing fatigue is a factor that encourages people to exercise [32]. Therefore, more intervals and a lesser exercise duration can be designed to improve exercise adherence. In reality, individuals prefer to have short breaks from prolonged exercise, which will enhance their persistence. Therefore, the findings of this study may thus support the use of intermittent exercise to reduce arterial stiffness in people.

Future research may confirm these results in order to successfully enhance cardiovascular health and lower the risks related to sedentary behavior through long-term lifestyle changes. In addition, future studies can explore potential mechanisms involving the effects of longer intervals of intermittent exercise on arterial stiffness.

This research has some limitations. For starters, we did not conduct a mechanistic study to investigate the mechanism. Second, the study only included healthy young people; no additional research was conducted on other populations, such as the elderly or patients with chronic diseases. Third, this study focused on acute exercise and did not include a long-term intervention exercise study.

## 5. Conclusions

The main findings of this study are that acute moderately mild exercise transiently reduces human arterial stiffness. Furthermore, acute moderately mild intermittent exercise has a superior effect compared to moderate continuous exercise. The arterial stiffness was improved by shorter durations of intermittent activity, and this improvement could be further increased by adding more exercise segments.

The greater impact of intermittent exercise over continuous exercise on arterial stiffness was unaffected by the decrease in overall exercise volume. The findings of this research suggest that the intervals of intermittent exercise and the number of exercise intervals play more significant roles in modulating arterial stiffness than the total amount of exercise.

## Figures and Tables

**Figure 1 ijerph-19-16847-f001:**
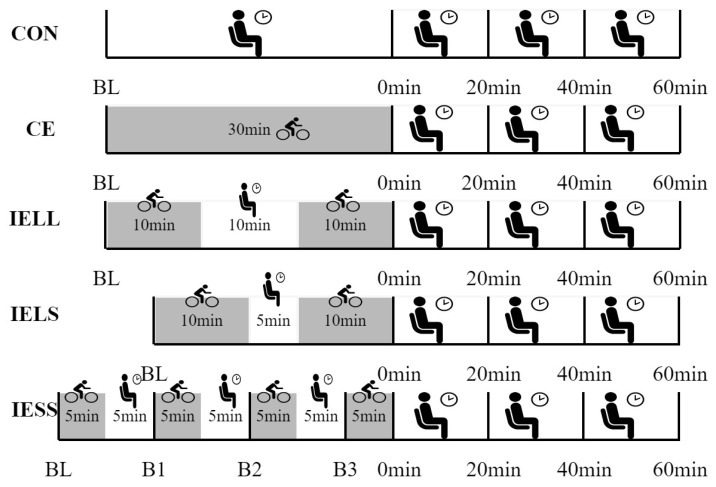
Exercise time, intervals, and measurement study methods for five trials: CON (control trial), CE (one 30 min session), IELL (two repeats of 10 min cycling sessions with a 10 min pause between sessions), IELS (two repeats of 10 min cycling sessions with a 5 min pause between sessions), and IESS (four repeats of 5 min cycling sessions with a 5 min pause between sessions). In CE, IELL, IELS. and IESS tests, the measured time points were baseline (BL), immediate (0 min), 20 min, 40 min, and 60 min after exercise. Additionally, three additional baPWV measures were taken shortly before the IESS trial’s second (B1), third (B2), and fourth 5 min bouts.

**Figure 2 ijerph-19-16847-f002:**
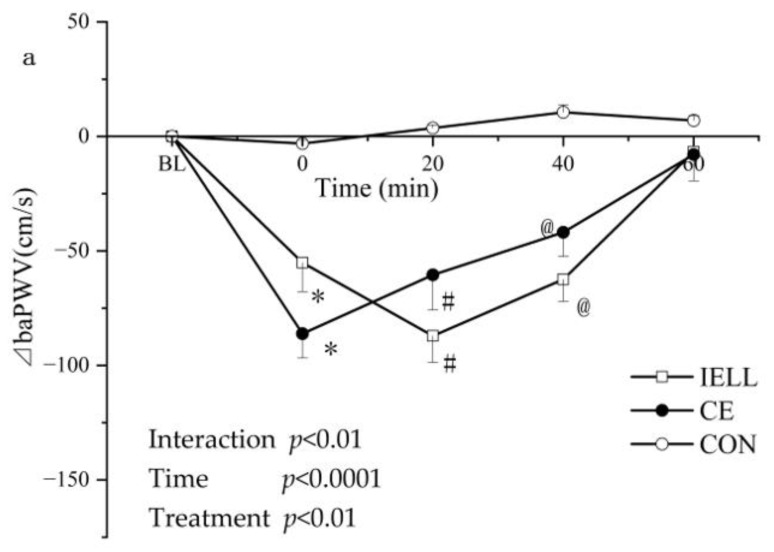
Exercise-induced arterial stiffness response time series with and without. The mean (±SD) is time-dependent ⊿baPWV changes in CON, CE, and IELL trials (**a**) and CON, CE, and IELS trials (**b**). * *p* < 0.001, CE, IELL, and IELS vs. CON at 0 min. # *p* < 0.05, CE, IELL, and IELS vs. CON at 20 min. @ *p* < 0.05, CE, IELL, and IELS vs. CON at 40 min. & *p* < 0.05, IELS vs. CON at 60 min.** *p* < 0.05, IELS vs. CE at 60 min. Data are the means ± SD, n = 20. CON (control trial), CE (one 30 min session), IELL (two repeats of 10 min cycling sessions with a 10 min pause between sessions), IELS (two repeats of 10 min cycling sessions with a 5 min pause between sessions), and IESS (four repeats of 5 min cycling sessions with a 5 min pause between sessions).

**Figure 3 ijerph-19-16847-f003:**
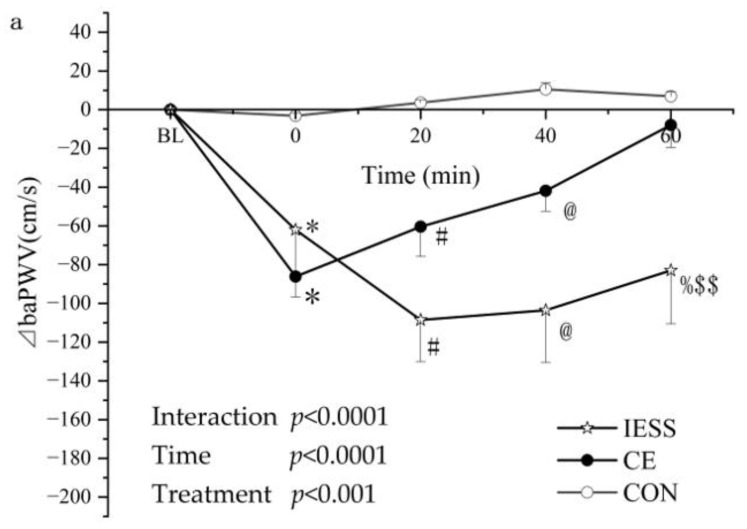
Exercise-induced arterial stiffness time courses with various sessions and brief intersessions. The mean (±SD) ⊿baPWV changes in CON, CE. and IESS trials (**a**) and CON, IELS, and IESS trials (**b**). * *p* < 0. 001, CE, IELL, IELS, and IESS vs. CON at 0 min. # *p* < 0.05, CE, IELL, and IELS vs. CON at 20 min. @ *p* < 0.05, CE, IELL, and IELS vs. CON at 40 min. & *p* < 0.05, IELS vs. CON at 60 min. % *p* < 0.05, IELS vs. CON at 60 min. $ *p* < 0.05, IESS vs. IELS at 60 min. $$ *p* < 0.05, IESS vs. CE at 60 min. Data are the means ± SD, n = 20. CON (control trial), CE (one 30 min session), IELL (two repeats of 10 min cycling sessions with a 10 min pause between sessions), IELS (two repeats of 10 min cycling sessions with a 5 min pause between sessions), and IESS (four repeats of 5 min cycling sessions with a 5 min pause between sessions).

**Figure 4 ijerph-19-16847-f004:**
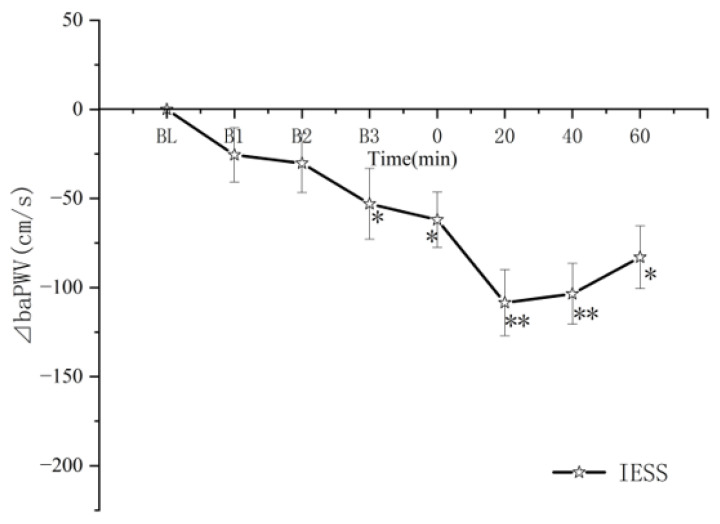
Mean (±SD) time-dependent ⊿baPWV changes in IESS trials. * *p* < 0. 001 vs. BL; ** *p* < 0.001 vs. BL. Data are the means ± SD, n = 20. CON (control trial), CE (one 30 min session), IELL (two repeats of 10 min cycling sessions with a 10 min pause between sessions), IELS (two repeats of 10 min cycling sessions with a 5 min pause between sessions), and IESS (four repeats of 5 min cycling sessions with a 5 min pause between sessions).

**Table 1 ijerph-19-16847-t001:** Subject characteristics (n = 20).

	Value (Mean ± SE)
Age (years)	20.4 ± 0.4
Height (cm)	178.5 ± 2.1
Weight (kg)	68.5 ± 4.5
BMI (kg/m^2^)	23.1 ± 3.3
Rest heart rate (beats/min)	62.3 ± 0.9
Systolic BP (mmHg)	115.9 ± 8.3
Diastolic BP (mmHg)	61.5 ± 5.1
baPWV (cm/s)	1028.3 ± 105.5

Values are means ± SE. BMI, body mass index; BP, blood pressure.

## Data Availability

Not applicable.

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
