# Peer review of "Acute Effects of the Interval and Duration of Intermittent Exercise on Arterial Stiffness in Young Men"

_ijerph, 2022, doi:10.3390/ijerph192416847_

Round 1
Reviewer 1 Report
Dear authors
I was delighted reviewing your manuscript. Here my comments to improve the overall quality of your paper:
Introduction: please revise line 35-36 to be clearer. overall, this section needs to be revised to improve the flow of the information and increase clarity of research gap, objective, and significance
results: there is no need to include table 2 and its related results
in the discussion section I would like if you included a paragraph about the limitation of the current study
Thank you
Author Response
Dear Editors and Reviewers:
Thank you for your letter and for the reviewers’ comments concerning our manuscript entitled “Acute Effects of Interval and Duration of Intermittent Exercise on Arterial Stiffness in young men” (ID:ID: ijerph-2072243). Those comments are all valuable and very helpful for revising and improving our paper, as well as the important guiding significance to our research. We have studied the comments carefully and made corrections, which we hope meet with approval. Revised portions are marked in red in the paper. The main corrections in the paper and the responses to the reviewer’s comments are as flowing:
Responds to the reviewer’s comments:
Point 1: Introduction: please revise line 35-36 to be clearer. overall, this section needs to be revised to improve the flow of the information and increase clarity of research gap, objective, and significance
Response 1:
â‘ As the Reviewer suggested that revise lines 35-36 be clearer.
In lines 35-36, the statements” In an environment of very low intensity and density of physical work, young people generally lack scientific exercise, but the atherosclerotic process is initiated at the stage of healthy youth[3]” were revised as lines 75-76 “A portion of young people lack the habit of exercise due to high work demands and limited time, but the atherosclerotic process starts during the stage of healthy adolescence[19].”
â‘¡In light of the Reviewer's suggestions, we have rewritten the introduction as follows to improve the flow of information and increase the clarity of the research gap, objective, and significance:
Introduction
Cardio-cerebrovascular disease (CVD), represented by strokes and heart attacks, is increasingly occurring in young people[1]. As a non-communicable chronic disease, early control of CVD risk factors can effectively prevent its occurrence. Arterial stiffness is valuable in predicting cardiovascular disease risk. Increased arterial stiffness is an independent cardiovascular disease risk factor, and physical activity may have a preventive effect on cardiovascular disease by improving arterial stiffness[2]. Several parameters can be used to evaluate the hardness of human arteries. For example, pulse wave velocity (PWV); an increase in PWV represents a decrease in vessel wall elasticity, distensibility, and compliance, and an increase in stiffness[3]. Cohort studies have demonstrated that aortic PWV predicts cardiovascular all-cause mortality[4]. Brachi-al-ankle pulse wave(baPWV) conduction velocity is a systemic arterial stiffness measurement method measured by analyzing brachial and tibial arterial waves[5]. BaPWV is considered the “gold standard” for evaluating arterial stiffness[6].According to Mitchell et al.[7], higher arterial stiffness is associated with an increased risk of first cardiovascular events. Over 10-12 years, a 1 m/s increase in baPWV was associated with a 0.9% absolute risk of stroke, 2.2% death, and 1.4% coronary heart disease. According to research, increased arterial stiffness in the elderly can harm brain health[8]. Arterial stiffness is caused by aging, hypertension, and inflammation[9].
Regular physical activity can improve health and prevent cardiovascular dis-ease[10]. Many studies have shown that exercise reduces cardiovascular risk, morbidity, and mortality[11, 12]. Aerobic exercise can help individuals improve their cardiorespiratory fitness and their vascular health. Increased aerobic capacity, for example, lowers blood pressure and arterial stiffness. Regular exercise improves arterial stiffness, long-distance running helps reduce arterial stiffness[13], even low-intensity resistance exercise helps reduce arterial stiffness[14], and high-intensity intermittent exercise helps reduce arterial stiffness in obese women[15]. People who exercise prefer moderate-intensity interval exercise to high-intensity continuous exercise and interval exercise because it is more acceptable and easier to stick to. The American College of Sports Medicine guidelines recommends continuous and intermittent exercise to stay healthy[16]. Regular exercise can improve arterial stiffness. For instance, Hu et al. [11]reported that after four weeks, the arterial stiffness of female university students decreased compared with that before moderate-intensity aerobic exercise training. Previous studies have shown intermittent and continuous exercise can improve arterial stiffness. Zheng et al. [17]showed that acute intermittent exercise is more beneficial than continuous exercise in reducing arterial stiffness for the same amount of exercise. However, the effect of intermittent exercise disappears when the intervals of intermittent exercise are too long[17], while the effect of intermittent exercise can be maintained by increasing the number of bouts of intermittent exercise[18]. These findings suggest that the duration and number of sessions affect the acute effects of intermittent exercise on arterial stiffness. Intermittent exercise is more acceptable to most people than continuous exercise. However, few studies have found the optimal interval duration and number of bouts.
A portion of young people lack the habit of exercise due to high work demands and limited time, but the atherosclerotic process starts during the stage of healthy ad-olescence[19]. Therefore, designing accessible and feasible exercise prescriptions to reduce arterial stiffness for an audience of healthy young adults is important to improve their vascular health and prevent CVD at an early stage. Hence, to investigate the superiority of intermittent exercise on the improvement of arterial stiffness, we hypothesized that intermittent exercise would have a better effect on arterial stiffness by shortening the duration of intervals and increasing the number of bouts.
Point 2: results: there is no need to include table 2 and its related results
Response 2: We removed table 2 and the results it included in response to the reviewer's suggestions. Lines 157-166 were deleted
Point 3:In the discussion section I would like if you included a paragraph about the limitation of the current study
Response 3: According to the reviewer's suggestions, we have added a section about the current study's limitations.
Lines 308–312 of a new paragraph are added as follows:
This research has some limitations. For starters, we did not conduct a mechanistic study to investigate the mechanism. Second, the study only included healthy young people; no additional research was conducted on other populations, such as the elderly or patients with chronic diseases. Third, this study focused on acute exercise and did not include a long-term intervention exercise study.
Special thanks to you for your good comments.

Reviewer 2 Report
This paper studiedthat intermittent exercise would have a better effect on arterial stiffness by shortening the duration of intervals and increasing the number of bouts. Becausedesigning accessible and feasible exercise prescriptions to reduce arterial stiffness for an audience of healthy young adults is important to improve their vascular health and prevent CVD at an early stage.
Major points:
1. The significance of this paper is dueto investigate the superiority of intermittent exercise on the improvement of arterial stiffness, we hypothesized that intermittent exercise would have a better effect on arterial stiffness by shortening the duration of intervals and increasing the number of bouts.
2. The purpose of the cross-self-controlled design is specific and scientific. The experiment was divided into five groups. The groups were clear, but the total number of people was small.The article explains the tests in detail, but does not explain the specific operation of the scale and the matters needing attention in the examination process, that is, a basic and common physical examinations.What assumptions and premises are the experiments and content of the article based on? Another point to be mentioned is the experimental equipment and equipment and personnel training and preparation process, these details should be paid more attention to.These are not elaborated in the article.
Minor points:
1. The chart explanations are professional, understandable, elegant and well explained. The introduction can be supplemented with more brief scientific study of the content and subject.Let the reader know more about the background and research status at home and abroad.
Author Response
Dear Editors and Reviewers:
Thank you for your letter and for the reviewers’ comments concerning our manuscript entitled “Acute Effects of Interval and Duration of Intermittent Exercise on Arterial Stiffness in young men” (ID:ID: ijerph-2072243). Those comments are all valuable and very helpful for revising and improving our paper, as well as the important guiding significance to our research. We have studied the comments carefully and made corrections, which we hope meet with approval. Revised portions are marked in red in the paper. The main corrections in the paper and the responses to the reviewer’s comments are as flowing:
Responds to the reviewer’s comments:
Point 1: The purpose of the cross-self-controlled design is specific and scientific. The experiment was divided into five groups. The groups were clear, but the total number of people was small. The article explains the tests in detail but does not explain the specific operation of the scale and the matters needing attention in the examination process, that is, a basic and common physical examinations. What assumptions and premises are the experiments and content of the article based on? Another point to be mentioned is the experimental equipment and equipment and personnel training and preparation process, these details should be paid more attention to. These are not elaborated in the article.
Response 1:
As the Reviewer suggested the specific operation of the scale and the matters needing attention in the examination process and the experimental equipment and equipment and personnel training and preparation process, these details should be paid more attention.
Line 88,” A medical health examination was completed by all subjects.” was added.
Line123-126,” The instrument operator was specially trained, and the subject lay flat on the test bed while sterile alcohol was wiped on the left upper arm, right upper arm, left ankle, and right ankle, and the appropriate wrist strap was fitted. ECG clips were placed on both the left and right wrists. “ was added.
Point 2: The chart explanations are professional, understandable, elegant and well explained. The introduction can be supplemented with more brief scientific study of the content and subject. Let the reader know more about the background and research status at home and abroad.
Response 2:
In light of the Reviewer's suggestions, we have rewritten the introduction as follows to supplement it with more brief scientific study of the content and subject. Let the reader know more about the background and research status at home and abroad:
Introduction
Cardio-cerebrovascular disease (CVD), represented by strokes and heart attacks, is increasingly occurring in young people[1]. As a non-communicable chronic disease, early control of CVD risk factors can effectively prevent its occurrence. Arterial stiffness is valuable in predicting cardiovascular disease risk. Increased arterial stiffness is an independent cardiovascular disease risk factor, and physical activity may have a preventive effect on cardiovascular disease by improving arterial stiffness[2]. Several parameters can be used to evaluate the hardness of human arteries. For example, pulse wave velocity (PWV); an increase in PWV represents a decrease in vessel wall elasticity, distensibility, and compliance, and an increase in stiffness[3]. Cohort studies have demonstrated that aortic PWV predicts cardiovascular all-cause mortality[4]. Brachi-al-ankle pulse wave(baPWV) conduction velocity is a systemic arterial stiffness measurement method measured by analyzing brachial and tibial arterial waves[5]. BaPWV is considered the “gold standard” for evaluating arterial stiffness[6].According to Mitchell et al.[7], higher arterial stiffness is associated with an increased risk of first cardiovascular events. Over 10-12 years, a 1 m/s increase in baPWV was associated with a 0.9% absolute risk of stroke, 2.2% death, and 1.4% coronary heart disease. According to research, increased arterial stiffness in the elderly can harm brain health[8]. Arterial stiffness is caused by aging, hypertension, and inflammation[9].
Regular physical activity can improve health and prevent cardiovascular dis-ease[10]. Many studies have shown that exercise reduces cardiovascular risk, morbidity, and mortality[11, 12]. Aerobic exercise can help individuals improve their cardiorespiratory fitness and their vascular health. Increased aerobic capacity, for example, lowers blood pressure and arterial stiffness. Regular exercise improves arterial stiffness, long-distance running helps reduce arterial stiffness[13], even low-intensity resistance exercise helps reduce arterial stiffness[14], and high-intensity intermittent exercise helps reduce arterial stiffness in obese women[15]. People who exercise prefer moderate-intensity interval exercise to high-intensity continuous exercise and interval exercise because it is more acceptable and easier to stick to. The American College of Sports Medicine guidelines recommends continuous and intermittent exercise to stay healthy[16]. Regular exercise can improve arterial stiffness. For instance, Hu et al. [11]reported that after four weeks, the arterial stiffness of female university students decreased compared with that before moderate-intensity aerobic exercise training. Previous studies have shown intermittent and continuous exercise can improve arterial stiffness. Zheng et al. [17]showed that acute intermittent exercise is more beneficial than continuous exercise in reducing arterial stiffness for the same amount of exercise. However, the effect of intermittent exercise disappears when the intervals of intermittent exercise are too long[17], while the effect of intermittent exercise can be maintained by increasing the number of bouts of intermittent exercise[18]. These findings suggest that the duration and number of sessions affect the acute effects of intermittent exercise on arterial stiffness. Intermittent exercise is more acceptable to most people than continuous exercise. However, few studies have found the optimal interval duration and number of bouts.
A portion of young people lack the habit of exercise due to high work demands and limited time, but the atherosclerotic process starts during the stage of healthy ad-olescence[19]. Therefore, designing accessible and feasible exercise prescriptions to reduce arterial stiffness for an audience of healthy young adults is important to improve their vascular health and prevent CVD at an early stage. Hence, to investigate the superiority of intermittent exercise on the improvement of arterial stiffness, we hypothesized that intermittent exercise would have a better effect on arterial stiffness by shortening the duration of intervals and increasing the number of bouts.
Special thanks to you for your good comments.
